# Pyriproxyfen-Treated Polypropylene Sheets and Resting Boxes for Controlling Mosquitoes in Livestock Operations

**DOI:** 10.3390/insects10020055

**Published:** 2019-02-18

**Authors:** Pattarapon Khemrattrakool, Jintana Yanola, Nongkran Lumjuan, Pradya Somboon

**Affiliations:** 1Center of Insect Vector Study, Department of Parasitology, Faculty of Medicine, Chiang Mai University, Chiang Mai 50200, Thailand; pattarapon.khem@gmail.com; 2Graduate PhD’s Degree Program in Parasitology, Faculty of Medicine, Chiang Mai University, Chiang Mai 50200, Thailand; 3Department of Medical Technology, Faculty of Associated Medical Sciences, Chiang Mai University, Chiang Mai 50200, Thailand; chintana.p@cmu.ac.th; 4Research Institute for Health Sciences, Chiang Mai University, Chiang Mai 50200, Thailand; nklumjuan@yahoo.com

**Keywords:** insect growth regulator, mosquitoes, pyriproxyfen, residual spray, resting box

## Abstract

Many insect vector species of medical and veterinary importance are found abundantly in areas where animals are held. In these areas, they often rest for a period of time on objects around the animals both before and after blood feeding. However, the use of neurotoxic insecticides for vector control is not advised for use in such shelters as these chemicals can pose hazards to animals. The present study evaluated the efficacy of pyriproxyfen (PPF), an insect growth regulator, applied to polypropylene sheets and resting boxes on the reproductivity of mosquitoes found in animal shelters in Chiang Mai, Thailand. The sheets sprayed with 666 mg PPF/m^2^ were set on the inner wall of a cowshed and kept in place for 3 h (6.00 to 9.00 pm). During this time, fully blood-fed female mosquitoes that landed and remained continuously on the sheets for 5, 10, and 20 min were collected. The results, involving *Anopheles subpictus*, *An. vagus*, *Culex gelidus*, *Cx. tritaeniorhynchus*, and *Cx. vishnui*, revealed significant reductions in oviposition rates, egg hatchability, pupation, and adult emergence in the PPF-treated groups compared to the control groups. Adult emergence rates were reduced to 85.6–94.9% and 95.5–100% in those exposed for 10 and 20 min, respectively. The sheets retained their effectiveness for three months. The PPF-treated (666 mg/m^2^) resting boxes (35 × 35 × 55 cm) were placed overnight at a chicken farm where *Cx. quinquefasciatus* predominated. Blood-fed mosquitoes were collected in the morning and reared in the laboratory. Oviposition rates were reduced by 71.7% and adult emergence was reduced by 97.8% compared to the controls. PPF residual spray on surface materials in animal sheds is a potential method for controlling mosquitoes. Further studies are needed to evaluate the impact of PPF-treated materials on wild populations.

## 1. Introduction

Many insect vector species of medical and veterinary importance are frequently found in animal sheds [1,2]. Moreover, they may acquire pathogens, such as viruses or protozoa, from animals and transfer these to humans during subsequent blood feedings [3,4]. Animal shelters are therefore important not only for sustaining insect vector populations, however also serve as potential sources of pathogens. Use of zooprophylaxis, which can reduce contact rates between vectors and humans by diverting bites towards non-human animals, can be integrated into vector control with chemical insecticides used in public health [5].

Many mosquito vector species readily feed on both humans and other animals, however some species are primarily zoophilic. For example, *Anopheles culicifacies* s.l., a malaria vector in India, feeds mainly on cattle [6]. Livestock serve as a plentiful source of blood which contributes to sustaining vector populations such as *An. arabiensis* and *An. funestus* s.l., the malaria vectors in Africa [7,8]. Many *Culex* species are vectors of arboviral diseases and filariasis [9]. Members of the *Culex vishnui* subgroup, such as *Cx. tritaeniorhynchus* and *Cx. vishnui*, are important vectors of Japanese encephalitis virus (JEV) and feed primarily on cattle and pigs [10]. *Culex quinquefasciatus*, a vector of bancroftian filariasis, feeds on a variety of hosts including birds, cattle, dogs, cats, and humans [11]. It is also abundant in pig farms and has been found to be infected with JEV [12], although its role in the transmission of the virus is not established.

Despite the risk of disease transmission, many prefer not to use neurotoxic insecticides to control insect vectors in animal sheds. Single dermal applications of recommended doses of pyrethroids resulted in the presence of residual insecticides in milk [13]. In addition, problems of insecticide resistance occurring in many mosquito species hampers the efficacy of insecticide usage [14]. To cope with these problems, the use of alternative insecticides that are environmentally friendly while also utilizing a different mode of action have been recommended for mosquito control [15].

Pyriproxyfen (PPF) is an insect growth regulator that has been used effectively to control mosquitoes in the aquatic stages [16,17]. Tarsal contact with high doses of PPF affects egg maturation in females and the hatchability of laid eggs [18,19,20,21,22]. In a semi-field study using small microcosms containing breeding habitats and a PPF-treated polyethylene bed net (350 mg/m^2^) with holes and a confined mouse inside, the population of *Ae. albopictus* in the microcosms was significantly reduced [23]. This reduction occurred because the female mosquitoes were exposed to PPF on the nets both before and after blood feeding during their attempts to enter and exit through the holes. Moreover, in an experimental hut, Ngufor et al. [24] demonstrated that bed nets treated with PPF and PPF + permethrin significantly reduced the fecundity and fertility of surviving blood-fed *An. gambiae*.

Artificial resting boxes of various sizes and shapes were used to collect mosquitoes that naturally prefer to rest in the shade or protected locations [1]. They have proved to be useful for quickly sampling *Ae. aegypti* inside houses and *Anopheles* populations outdoors [25,26]. Since PPF does not have a repellency effect [27], PPF-treated resting devices are effective against *Ae. aegypti* mosquitoes [28,29].

In animal shelters, we often observed that female mosquitoes may rest for a period of time on objects around animals. These resting periods occur both before and after blood feeding and may last for several hours. Some species, particularly after blood-feeding, may be found resting around animal sheds. This behavior allows us to develop new strategies for controlling zoophagic mosquito species.

## 2. Materials and Methods

### 2.1. Reproductive Effect of PPF on Mosquitoes Determined by Cone Bioassay

Since little is known about the effective dosage of PPF against *Culex* and *Anopheles* adult mosquitoes, our preliminary study was carried out by using CDC bottle bioassays [30] with 250 mL Duran bottles with inner surfaces of approximately 300 cm^2^. Fully blood-fed *Cx. quinquefasciatus* females (NIH strain, Bangkok, Thailand) were exposed for 30 min to varying doses of PPF. The results revealed that the oviposition of mosquitoes was totally inhibited at 333 mg/m^2^. However, the treatment of glass bottles with PPF is not directly applicable to surface treatment within animal shelters. In this experiment, we used a simple spraying method to coat a surface and then evaluated reductions in reproductivity of *Cx. quinquefasciatus* (NIH strain) and wild *Anopheles* and *Culex* females using cone bioassays according to WHO protocol [31]. For the laboratory *Cx. quinquefasciatus*, bioassays were performed on black polypropylene sheets (35 × 25 cm), while gray polypropylene sheets (65 × 122 cm) were used for bioassays with wild mosquitoes. The sheets were rubbed with sand paper and then sprayed with a PPF solution. This solution consisted of technical grade PPF dissolved in acetone at dosages of 333 mg/m^2^ and 666 mg/m^2^ and was administered using a foggy sprayer for ironing (1 L) (Moong Pattana PCL., Bangkok, Thailand). In a previous study, a dosage of 1.3% PPF (or about 455 mg/m^2^) was applied for the treatment of a barrier against *Ae. albopictus* [32]. An amount of 166 mL acetone was found to be sufficient to cover a 1 m^2^ area. Control sheets were sprayed with acetone alone. The sprayed sheets were allowed to dry and were then stored in a dark place until use.

For the bioassays, plastic cones were attached to each of polypropylene sheets and placed vertically (Figure 1). Then, 10 newly blood-fed *Cx. quinquefasciatus* females were released into each of the plastic cones and were exposed for 30 min. After exposure, the mosquitoes were transferred individually to plastic cups (236 mL), hereafter called ovicups, and were provided with a 10% sugar solution. For wild mosquitoes, fully blood-fed *Anopheles* and culicine mosquitoes were collected by the aid of an aspirator and touch from a cow shed at Ban Thung Kor Laan, Doi Saket District, Chiang Mai Province, Thailand. The collected mosquitoes were released into each cone (maximum 10 females) and were left for 30 min exposure. Thereafter, they were placed in cups and transferred to the laboratory. In the morning, all of the treated and control mosquitoes were identified morphologically using the keys of Rattanarithikul et al. [33,34] and were transferred individually to ovicups.

The exposed and control mosquitoes were reared for four days to ensure egg maturation. Three days after exposure, each cup was filled with 70 mL of hay infused water (10 g of dry grass submerged in 1 L of water for one week and diluted with 10 parts of water before use) as an oviposition medium. The females were allowed to lay eggs for a maximum of one week. The laid eggs were then allowed to hatch for three days. The hatched larvae in each ovicup were reared with rabbit food (Tops Rabbit^TM^, Nakhon Pathom, Thailand) until the emergence of adults. The oviposition rates (number of oviposited females per total females), numbers of eggs laid, eggs hatched, pupation rates, and adult emergence rates were recorded. The parental females were examined for insemination status and for the presence of retained eggs in the ovaries. The experiment was performed in triplicate for a total of 30 females in each group.

### 2.2. Reproductive Effects with Different Exposure Times

This experiment evaluated the effects on reproductivity in free-flying female mosquitoes that had rested continuously on PPF-treated sheets for various amounts of time. Four rubbed polypropylene sheets (65 × 122 cm, gray color) were sprayed with PPF at the dosage of 666 mg/m^2^, as mentioned previously. Control sheets were sprayed with acetone only. Thereafter, they were marked with grid lines and labeled (Figure 2) to facilitate locating the mosquitoes. The treated and control devices (four sheets each) were set up at dusk and allowed to stand for 3 h (6.00 to 9.00 pm) during which fully blood-fed females that remained continuously on the sheets for 5, 10, and 20 min were collected with the aid of an aspirator, torch, and timer. They were then transferred to the laboratory for rearing. In the laboratory, engorged female *Anopheles* and *Culex* were sorted out and morphologically identified to species. They were reared individually to determine the reproductive parameters as described previously.

### 2.3. Residual Activity of PPF-Treated Polypropylene Sheets

The above gray PPF-treated sheets (666 mg/m^2^) in Section 2.1 were kept in a dark place in the laboratory and their residual activity was evaluated 3 and 5 months after treatment. Engorged female mosquitoes were collected from the cowshed and transferred to the laboratory. After morphological identification, five blood-fed females of each mosquito species were released into the plastic cone which was attached to the PPF-treated polypropylene sheets for 30 min. Three replicates were conducted for a total of 15 females per species. After exposure, mosquitoes were kept individually in ovicups and were reared to determine the effects of PPF exposure on reproductivity, as mentioned previously.

### 2.4. PPF-Treated Resting Boxes for Controlling Cx. quinquefasciatus

This experiment was meant to evaluate the efficacy of PPF-treated resting boxes on the reproductivity of *Cx. quinquefasciatus* females. The experiment was conducted at a chicken farm (housing approximately 50 chickens) in Chiang Mai city. Mosquito resting boxes (35 × 35 × 55 cm) were constructed using black polypropylene sheets and had a circular hole (20 cm diameter) on top. The inner surface of the box was sprayed with technical grade PPF dissolved in acetone at a concentration of 666 mg/m^2^. The control box was sprayed with acetone alone.

One control and one PPF-treated resting box were placed at the same site on alternative nights for six consecutive nights, starting at 6 pm and collected at 8 am, in December 2016. All mosquitoes that were found in the box were removed and taken to the laboratory. Mosquitoes were identified to species according to Rattanarithikul et al. [33]. Fully engorged *Cx. quinquefasciatus* females were selected at random and reared individually in ovicups. Three days after, each cup was filled with the oviposition medium and mosquitoes were allowed to lay eggs for a maximum of one week. The oviposition rate, number of laid eggs, hatched larvae, pupae, and adults were recorded. Ovaries were dissected and examined for insemination status and the presence of retained eggs.

The PPF-treated and control boxed were stored in a dark room in the laboratory and reused on months 3 and 5 to observe the residual activity. Collection, rearing, and the determination of reproductive parameters were as described previously.

The potential attraction of adult *Cx. quinquefasciatus* to the resting boxes was evaluated in February 2017. In this study, we wanted to compare the number of mosquitoes collected by one box and two boxes which were placed on alternative nights for six consecutive nights. The station of one box was fixed and the second box was placed approximately 15 m apart. All *Cx. quinquefasciatus* found in the boxes were separated, counted, and scored as non-blood fed, blood-fed, half-gravid, and gravid. Male mosquitoes were also counted.

### 2.5. Statistical Analysis

The oviposition rates were analyzed by Pearson’s chi-squared test or Fisher’s Exact Test. Mean numbers of eggs laid, hatched larvae, pupae, and emergent adults were analyzed by negative binomial regression using SPSS version 22.0 (IBM Corp., Armonk, NY, USA).

## 3. Results

### 3.1. Reproductive Effect of PPF on Mosquitoes Determined by Cone Bioassay

The effects of tarsal contact with PPF were evaluated in the laboratory *Cx. quinquefasciatus* and four species of wild mosquitoes, including *An. subpictus*, *An. vagus*, *Cx. tritaeniorhynchus*, and *Cx. vishnui*. PPF concentration had a strong effect on the oviposition of all species, with 100% suppression in the 666 mg/m^2^ groups (Table 1). In the 333 mg/m^2^ groups, 84.7–88.9% reductions in the oviposition rate compared with the controls were observed. Mean numbers of laid eggs were reduced by 88.2–92.5%. Most of the hatched larvae developed to pupae and adults. Ultimately, the adult emergence was reduced by 87.6–94.2% compared with the control groups.

### 3.2. Reproductive Effects with Different Exposure Times

The reproductive effects of PPF-treated polypropylene sheets on free-flying engorged female mosquitoes, including *An. subpictus*, *An. vagus*, *Cx. gelidus*, *Cx. tritaeniorhynchus*, and *Cx. vishnui*, with different exposure times were evaluated. It was observed that the density of mosquitoes resting on the PPF-treated sheets was more or less the same as found on the control sheets, as well as other objects in the surroundings. Exposure for 5 min resulted in a moderate reduction in the oviposition rate in all species, except that of *An. vagus* which was not significantly different from the control (Table 2). More effect was observed in *Culex* (35.8–62.7% reduction) than *Anopheles* species (7.7–33.3% reduction). The mean numbers of laid eggs of *Culex* and *Anopheles* species were reduced 31.9–69.4% and 23.7–27.6%, respectively. Exposure for 10 min largely reduced the oviposition rate (74.4–89.8% reduction) and mean numbers of laid eggs (83.3–93.0% reduction) in all species. Exposure for 20 min completely inhibited oviposition in *Cx. tritaeniorhynchus* and *Cx. vishnui*, whereas one in five *An. vagus* laid eggs. At this exposure time, no observation could be made for *An. subpictus* and *Cx. gelidus* due to insufficient numbers. Most of the laid eggs were hatchable to be larvae and low mortality was observed until adult emergence. Ultimately, adult emergence rates of 5-min exposed *Anopheles* and *Culex* species were reduced by 16.1–23.0% and 32.3–69.8%, respectively. Exposure for 10 and 20 min largely suppressed the adult emergence rates (85.6–94.9% and 95.5–100% reductions, respectively) in all the species that were studied.

### 3.3. Residual Activity of PPF-Treated Polypropylene Sheet

The residual effects of PPF-treated polypropylene sheet (666 mg/m^2^) were evaluated by the cone bioassays with 30 min exposure three and five months after treatment (data for the first month can be found in Table 1). In this study, four species of mosquitoes, *An. subpictus*, *An. vagus*, *Cx. tritaeniorhynchus*, and *Cx. vishnui*, were evaluated (Table 3). In the first month, 86.7–90.0% of females in the control groups laid eggs, whereas none of the females in the exposed groups produced eggs. Three months after the initial treatment, the oviposition rates, mean numbers of laid eggs, and adult emergence among the exposure groups were reduced by 50.0–66.7%, 58.1–69.6%, and 61.6–71.0%, respectively, when compared with the control groups of their respective species (*p* < 0.05). After five months, 80–86.7% of females in the control groups laid eggs compared with 46.7–60% in the exposed groups. However, these oviposition rates are not significantly different (*p* > 0.05), except for *Cx. tritaeniorhynchus*. Egg production was reduced in all species, however this was not statistically different from the control groups. The effects on hatchability and pupation towards adult emergence were not significantly reduced, except for *Cx. tritaeniorhynchus* (44.9%, *p* < 0.05) in the PPF-treated groups.

### 3.4. PPF-Treated Resting Box for Controlling Cx. quinquefasciatus

In this experiment, the efficacy of PPF-treated resting boxes (666 mg/m^2^) on the reproductivity of the feral *Cx. quinquefasciatus* females was evaluated. Significant differences in oviposition rates were observed between the control and PPF-treated groups (Table 4). Approximately 90% of females in the control group laid eggs, whereas only 17.7% of females in the PPF-treated group laid eggs, a 71.7% reduction. Moreover, the mean numbers of larvae, pupae, and emergent adults were all significantly lower than the control groups (*p* < 0.001). Adult emergence was reduced by 97.8% compared with the control groups. After three months, the PPF-treated resting box was still highly effective in reducing the reproductivity of *Cx. quinquefasciatus*, with 99.3% reduction in adult emergence (*p* < 0.001). After five months, efficacy declined, however all of the parameters were still significantly reduced compared to the control (*p* < 0.001). The adult emergence rate was reduced by approximately 60%.

*Culex quinquefasciatus* found in the resting boxes without PPF consisted of non-blood fed, blood-fed, half-gravid, and gravid female and male mosquitoes (Table 5), indicating that the boxes were attractive to both sexes and all physiological stages of female mosquitoes. Overall, the majority of mosquitoes found in the boxes were blood-fed females (51.9%), followed by males (22.9%). The numbers of mosquitoes were approximately double when two boxes were placed. However, the mean numbers of mosquitoes per box were not significantly different regardless of whether one or two boxes were placed (*p* > 0.05).

## 4. Discussion

Most studies on the use of PPF-treated surfaces for mosquito control have focused on the development of PPF-treated ovitraps which are suitable for container-breeding mosquitoes such as *Aedes aegypti* and *Ae. albopictus* [18,19,35,36,37,38,39,40,41,42,43,44]. However, ovitraps are generally not attractive for most *Culex* or *Anopheles* mosquitoes [45]. Cattle, pigs, and chicken are commonly found throughout rural Thailand. These animals serve as important blood sources for both vector and non-vector mosquitoes. To our knowledge, our study appears to be the first to demonstrate that residual PPF spraying on a small surface area is a potential method for controlling *Culex* and *Anopheles* mosquitoes when they are seeking blood meals in animal shelters.

In the current study, exposure of *Culex* and *Anopheles* mosquitoes for 30 min on the PPF-sprayed polypropylene sheets at a dosage of 666 mg/m^2^ totally disrupted the fecundity of these mosquitoes. This concentration is higher than that tested in our previous study using the CDC bottle assay in which the oviposition of *Cx. quinquefasciatus* mosquitoes was totally inhibited at 333 mg/m^2^. This difference may be attributed to there being a small loss of chemicals during spraying and the fact that the PPF concentration over the sprayed sheet was not homogenized. Nonetheless, our spraying method is a simple and inexpensive method for applying PPF to surfaces and could be repeated easily. Therefore, this concentration (666 mg/m^2^) is recommended for residual spraying. This concentration is higher than the dose of 1% PPF (350 mg/m^2^) that is normally used for coating nets, including commercially available treated nets (e.g., Olyset Duo^®^, Sumitomo) [22,23,24,46,47]. Higher doses of PPF (≥ 1000 mg/m^2^) have also been evaluated [18,35], however such doses may be too high and not economical for large scale use.

The reproductive effect of PPF varies depending on mosquito species. In pyrethroid-resistant *Ae. aegypti* strains, exposure for 10 min on PPF-sprayed sheets at a dosage of 333 mg/m^2^ completely inhibited fecundity [28]. In the current study, *Culex* species appeared to be more sensitive than *Anopheles* species studied, particularly when the exposure time was short (5 min). Such differences are probably due to differences in PPF absorption, resistance levels, and/or resistance mechanisms. Insecticide resistance with various mechanisms has been detected in several species of *Culex*, *Aedes*, and *Anopheles* in Thailand [48,49]. A recent study demonstrated that a subset of cytochrome P450 enzymes can metabolize PPF in *An. gambiae* [50]. However, the current study suggests that the effects of long PPF exposure (20 min or longer) are not mitigated by currently known resistance mechanisms.

The residual effect of PPF-sprayed sheets or resting boxes, which was effective over three months, is based on 30 min of exposure time. This may underestimate what would occur in natural conditions because it was observed that newly blood-fed females usually rest for several hours on surrounding surfaces in cowsheds before they disappeared in the morning. Newly blood-fed females are likely to rest in their preferable resting places, including the resting box, one day or longer during egg maturation. Longer exposures to PPF could increase the chances of chemical uptake. However, PPF is light-sensitive and this property may be disadvantageous for outdoor residual spraying applications. In the current study, the PPF-treated sheets were kept in a dark place during the evaluation of the residual effect. Hence, shorter periods of residual activity is expected if it is used under real conditions. This would need to be evaluated in field conditions.

Besides the polypropylene sheets that were used in the current study, PPF may be applied to a variety of surface substrates. In general, insecticidal residues are more persistent on nonporous surfaces such as metal and tile compared with more porous surfaces such as concrete or wood [51]. Little is known about the persistence of PPF on different surfaces under natural conditions and, as such, warrants further study. In addition, some cattle owners protect their animals from insect bites by using nets. Treatment of nets with PPF may be an alternative for controlling insect populations.

The current study demonstrated that the artificial resting boxes are attractive to various physiological stages of *Cx. quinquefasciatus* females, mainly blood-fed, as well as males. This species is active at nighttime and at dawn seeks rest in dark corners in and around homes. Hence, treatment with residual PPF could be useful for controlling this vector species, especially in areas where they are found in abundance, such as chicken farms or animal sheds. PPF affects the fecundity of female mosquitoes, including those exposed before or after blood-feeding or while gravid [18,19,22,24]. In addition, our device could also contaminate male mosquitoes and may help to transfer PPF to virgin females via mating, as observed in *Ae. albopictus* and *Ae. aegypti* [28,38].

Both the PPF-treated sheets and PPF-treated resting boxes represent a “passive” approach to the control of mosquitoes in livestock. Such methods do not require much attention yet continue to affect mosquitoes over time. Retreatment may be applied every three months to maintain control. The portable nature of the polypropylene sheets and the resting boxes as well as their low cost per unit suggests that they could be efficiently deployed in high numbers with minimal disruption to animals or local residents. The results from this study provide important information on the use of PPF which could be implemented in the future for controlling mosquito-borne diseases.

## 5. Conclusions

The use of PPF-treated sheets and resting boxes may enhance the effectiveness of controlling *Anopheles* and *Culex* mosquitoes in livestock. Further evaluations of these devices under natural field conditions are needed to determine their efficacy for controlling mosquito populations.

## Figures and Tables

**Figure 1 insects-10-00055-f001:**
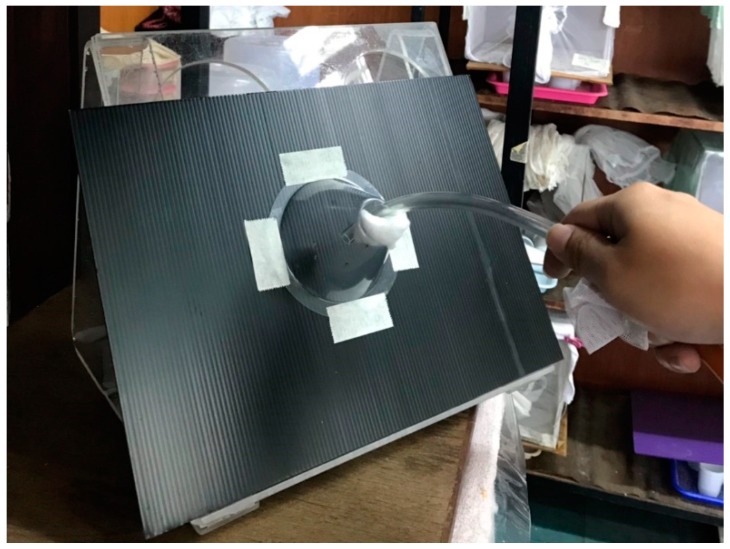
Cone bioassay.

**Figure 2 insects-10-00055-f002:**
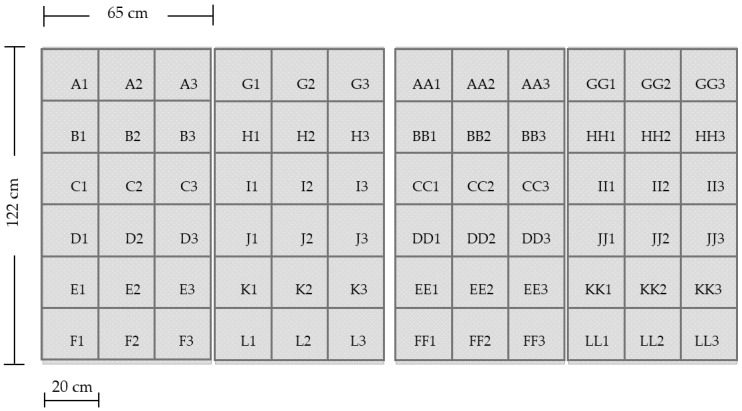
Polypropylene sheets with grid lines and labeling numbers.

**Table 1 insects-10-00055-t001:** Reproductivity of engorged females after 30 min of exposure to PPF (pyriproxyfen)-treated polypropylene sheets in the cone bioassay.

Species	PPF (mg/m^2^)	Percent Female Ovipositing (n = 30)	Eggs/Female (Mean ± S.D.)	Larvae/Female (Mean ± S.D.)	Pupae/Female (Mean ± S.D.)	Adult/Female (Mean ± S.D.)
*An. subpictus*	Control	86.7 ^a^	119.2 ± 48.8 ^a^	107.6 ± 44.0 ^a^	99.0 ± 40.6 ^a^	96.3 ± 39.4 ^a^
	333	13.3 ^b^	14.1 ± 37.0 ^b^	12.8 ± 33.5 ^b^	12.1 ± 31.6 ^b^	11.9 ± 30.9 ^b^
	666	0 ^b^	0 ^c^	0 ^c^	0 ^c^	0 ^c^
*An. vagus*	Control	86.7 ^a^	127.8 ± 52.1 ^a^	119.9 ± 48.9 ^a^	110.7 ± 45.3 ^a^	107.4 ± 43.9 ^a^
	333	13.3 ^b^	11.4 ± 30.2 ^b^	10.4 ± 27.3 ^b^	10.0 ± 26.2 ^b^	9.9 ± 25.8 ^b^
	666	0 ^b^	0 ^c^	0 ^c^	0 ^c^	0 ^c^
*Cx.*	Control	100 ^a^	137.7 ± 12.9 ^a^	118.6 ± 11.1 ^a^	109.7 ± 10.2 ^a^	99.7 ± 9.3 ^a^
*quinquefasciatus*	333	13.3 ^b^	10.3 ± 27.1 ^b^	7.0 ± 18.2 ^b^	6.5 ± 17.1 ^b^	5.8 ± 15.1 ^b^
	666	0 ^b^	0 ^c^	0 ^c^	0 ^c^	0 ^c^
*Cx.*	Control	90 ^a^	115.4 ± 40.5 ^a^	109.5 ± 38.6 ^a^	103.2 ± 36.7 ^a^	99.7 ± 35.1 ^a^
*tritaeniorhynchus*	333	10 ^b^	9.8 ± 30.0 ^b^	9.0 ± 27.5 ^b^	8.6 ± 26.3 ^b^	8.3 ± 25.3 ^b^
	666	0 ^b^	0 ^c^	0 ^c^	0 ^c^	0 ^c^
*Cx. vishuni*	Control	90 ^a^	112.4 ± 39.6 ^a^	104.8 ± 37.0 ^a^	94.3 ± 33.4 ^a^	92.0 ± 32.5 ^a^
	333	13.3 ^b^	11.1 ± 29.3 ^b^	10.2 ± 26.8 ^b^	9.8 ± 25.8 ^b^	9.7 ± 25.4 ^b^
	666	0 ^b^	0 ^c^	0 ^c^	0 ^c^	0 ^c^

In each species, different letters (a, b, c) in the same columns indicate statistical differences (*p* < 0.05).

**Table 2 insects-10-00055-t002:** Reproductivity of naturally engorged female mosquitoes after, 5, 10, and 20 min exposure to PPF-treated polypropylene sheets (666 mg/m^2^).

Species	Exposure time (min)	Female Ovipositing (n, %)	Eggs/Female (Mean ± S.D.)	Larvae/Female (Mean ± S.D.)	Pupae/Female (Mean ± S.D.)	Adult/Female (Mean ± S.D.)
*An. subpictus*	Control	30, 100 ^a^	137.5 ± 22.8 ^a^	118.0 ± 19.0 ^a^	107.8 ± 17.4 ^a^	104.8 ± 16.9 ^a^
	5	24, 66.7 ^b^	99.5 ± 73.1 ^a^	90.2 ± 66.2 ^a^	84.7 ± 62.3 ^a^	80.7 ± 59.2 ^a^
	10	20, 20.0 ^c^	17.2 ± 35.6 ^b^	14.0 ± 28.9 ^b^	13.0 ± 26.8 ^b^	12.6 ± 26.0 ^b^
	20	nd	-	-	-	-
*An. vagus*	Control	30, 86.7 ^a^	135 ± 56.0 ^a^	118.5 ± 49.1 ^a^	112.3 ± 46.3 ^a^	103.2 ± 42.5 ^a^
	5	20, 80.0 ^a^	103.6 ± 54.5 ^a^	96.0 ± 50.5 ^a^	90.2 ± 47.4 ^a^	86.6 ± 45.5 ^a^
	10	18, 22.2 ^b^	19.6 ± 37.9 ^b^	12.4 ± 24.0 ^b^	11.3 ± 21.9 ^b^	11.1 ± 21.4 ^b^
	20	5, 20.0 ^b^	16.8 ± 37.6 ^b^	5.0 ± 11.2 ^b^	4.8 ± 10.7 ^b^	4.6 ± 10.3 ^b^
*Cx. gelidus*	Control	28, 89.3 ^a^	120.1 ± 45.9 ^a^	107.0 ± 41.4 ^a^	102.1 ± 40.2 ^a^	97.6 ± 39.0 ^a^
	5	24, 45.8 ^b^	43.5 ± 54.2 ^b^	36.0 ± 45.5 ^b^	33.3 ± 42.8 ^b^	31.8 ± 41.2 ^b^
	10	20, 10.0 ^c^	8.4 ± 26.0 ^c^	5.9 ± 18.3 ^c^	5.3 ± 16.5 ^c^	5.0 ± 15.4 ^c^
	20	nd	-	-	-	-
*Cx.*	Control	28, 85.7 ^a^	103.5 ± 45.9 ^a^	92.9 ± 41.9 ^a^	88.2 ± 40.2 ^a^	84.0 ± 38.6 ^a^
*tritaeniorhynchus*	5	20, 55.0 ^b^	71.5 ± 68.0 ^a^	64.8 ± 61.7 ^a^	60.4 ± 57.8 ^a^	56.9 ± 54.7 ^a^
	10	18, 16.7 ^c^	17.3 ± 40.1 ^b^	15.0 ± 34.6 ^b^	13.7 ± 31.6 ^b^	12.1 ± 27.8 ^b^
	20	4, 0.0 ^c^	0 ^c^	0 ^c^	0^c^	0 ^c^
*Cx. vishuni*	Control	28, 89.3 ^a^	116.9 ± 44.3 ^a^	101.7 ± 38.6 ^a^	97.3 ± 37.0 ^a^	94.5 ± 36.0 ^a^
	5	18, 33.3 ^b^	35.8 ± 52.6 ^b^	31.9 ± 47.0 ^b^	29.6 ± 43.4 ^b^	28.5 ± 41.9 ^b^
	10	25, 16.0 ^b^	13.8 ± 33.2 ^b^	9.8 ± 23.7 ^c^	9.0 ± 21.5 ^c^	7.9 ± 19.0 ^c^
	20	6, 0.0 ^c^	0 ^c^	0 ^c^	0 ^c^	0 ^c^

In each species, different letters (a, b, c) in the same columns indicate statistical differences (*p* < 0.05), nd: not determined.

**Table 3 insects-10-00055-t003:** Residual effects of PPF-treated polypropylene sheets (666 mg/m^2^) on the reproductivity of engorged females after 30 min of exposure in the cone bioassays.

Species	Group	Female Ovipositing (n, %)	Eggs/Female (Mean ± S.D.)	Larvae/Female (Mean ± S.D.)	Pupae/Female (Mean ± S.D.)	Adult/Female (Mean ± S.D.)
		**Three months after treatment**				
*An. subpictus*	Control	15, 80.0	111.0 ± 58.1 ^a^	102.3 ± 53.6 ^a^	95.7 ± 50.2^a^	93.3 ± 48.9 ^a^
	PPF	15, 26.7	33.7 ± 59.3 ^b^	30.2 ± 53.6 ^b^	27.9 ± 49.3 ^b^	27.1 ± 47.8 ^b^
*An. vagus*	Control	15, 80.0	110.1 ± 57.5 ^a^	103.5 ± 54.1 ^a^	96.9 ± 50.8 ^a^	94.6 ± 49.5 ^a^
	PPF	15, 40.0	46.1 ± 60.5 ^b^	40.9 ± 53.7 ^b^	37.7 ± 49.2 ^b^	36.3 ± 47.3 ^b^
*Cx. tritaeniorhynchus*	Control	15, 86.7	108.8 ± 45.6 ^a^	97.2 ± 41.1 ^a^	91.2 ± 38.4 ^a^	90.1 ± 37.9 ^a^
	PPF	15, 33.3	40.0 ± 59.6 ^b^	31.9 ± 47.3 ^b^	29.8 ± 44.0 ^b^	28.3 ± 41.7 ^b^
*Cx. vishnui*	Control	15, 80.0	98.3 ± 51.8 ^a^	90.1 ± 47.5 ^a^	83.9 ± 44.3 ^a^	82.3 ± 43.3 ^a^
	PPF	15, 33.3	37.9 ± 55.8 ^b^	33.5 ± 49.3 ^b^	31.3 ± 46.1 ^b^	30.2 ± 44.4 ^b^
		**Five months after treatment**				
*An. subpictus*	Control	15, 86.7	117.9 ± 49.2 ^a^	108.5 ± 45.3 ^a^	96.5 ± 40.3 ^a^	94.3 ± 39.1 ^a^
	PPF	15, 60.0	80.5 ± 68.5 ^a^	71.3 ± 60.9 ^a^	65.7 ± 56.0 ^a^	62.9 ± 53.5 ^a^
*An. vagus*	Control	15, 80.0	110.9 ± 58.3 ^a^	102.4 ± 53.9 ^a^	94.9 ± 49.9 ^a^	92.9 ± 48.8 ^a^
	PPF	15, 46.7	64.0 ± 71.1 ^a^	57.5 ± 56.5 ^a^	52.7 ± 58.5 ^a^	50.1 ± 55.6 ^a^
*Cx. tritaeniorhynchus*	Control	15, 86.7	108.6 ± 45.3 ^a^	100.6 ± 41.8 ^a^	91.9 ± 38.2 ^a^	90.3 ± 37.5 ^a^
	PPF	15, 53.3	66.1 ± 64.5 ^a^	57.2 ± 55.8 ^b^	52.3 ± 51.0 ^b^	49.8 ± 48.6 ^b^
*Cx. vishnui*	Control	15, 80.0	101.9 ± 53.8 ^a^	94.1 ± 49.6 ^a^	84.6 ± 44.6 ^a^	82.8 ± 43.5 ^a^
	PPF	15, 53.3	65.7 ± 64.4 ^a^	56.7 ± 55.9 ^a^	52.9 ± 52.0 ^a^	50.6 ± 49.7 ^a^

Different letters (a, b, c) in the same columns between the control and PPF groups indicates statistical differences (*p*
*<* 0.05).

**Table 4 insects-10-00055-t004:** Effects of the PPF-treated resting box (666 mg/m^2^) on the reproductivity of engorged *Cx. quinquefasciatus* and residual effects after three and five months of treatment.

Group	Female Ovipositing (n, %)	Eggs/Female (Mean ± S.D.)	Larvae/Female (Mean ± S.D.)	Pupae/Female (Mean ± S.D.)	Adult/Female (Mean ± S.D.)	Emergence Reduction (%)
**First month**						
Control	66, 89.4	132.3 ± 47.5 ^a^	121.3 ± 44.8 ^a^	113.5 ± 42.0 ^a^	110.9 ± 41.1 ^a^	
PPF	79, 17.7	11.2 ± 24.7 ^b^	3.0 ± 7.7 ^b^	2.7 ± 6.8 ^b^	2.5 ± 6.4 ^b^	97.8
**Three months after treatment**						
Control	30, 100	139.5 ± 11.9 ^a^	131.5 ± 11.4 ^a^	122.8 ± 10.7 ^a^	118.5 ± 11.1 ^a^	
PPF	30, 13.3	7.0 ± 18.4 ^b^	0.9 ± 2.4 ^b^	0.9 ± 2.4 ^b^	0.9 ± 2.4 ^b^	99.3
**Five months after treatment**						
Control	30, 100	147.6 ± 11.0 ^a^	136.4 ± 11.1 ^a^	127.0 ± 10.6 ^a^	124.7 ± 11.1 ^a^	
PPF	30, 46.7	68.2 ± 74.4 ^b^	61.6 ± 67.2 ^b^	55.0 ± 60.0 ^b^	50.3 ± 54.7 ^b^	59.7

Different letters (a, b, c) in the same columns between the control and PPF groups indicates statistical differences (*p <* 0.05).

**Table 5 insects-10-00055-t005:** Numbers of *Cx. quinquefasciatus* found in one and two resting boxes without PPF.

Number of Resting Box	Female	Male(%)	Total	Average/Box/Night
Non-Blood Fed(%)	Blood-Fed(%)	Half-Gravid(%)	Gravid(%)
1	193(13.5)	658(46.0)	131(9.2)	60(4.2)	388(27.1)	1430	476.7
2	297(11.5)	1427(55.1)	203(7.8)	129(5.0)	533(20.6)	2589	431.5
Total	490(12.2)	2058(51.9)	334(8.3)	189(4.7)	921(22.9)	4019

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
