# Peer review of "Pyriproxyfen-Treated Polypropylene Sheets and Resting Boxes for Controlling Mosquitoes in Livestock Operations"

_insects, 2019, doi:10.3390/insects10020055_

Round 1

Reviewer 1 Report

I have reviewed the manuscript titled “Pyriproxyfen-treated polypropylene sheets and resting boxes for controlling misquotes found in cowshed and chicken farm” by Khemrattrakool et al.

 The manuscript reports on a series of laboratory and field experiments designed to test the potential effectiveness of pyriproxyfen based mosquito control strategies in association with livestock facilities.

The manuscript is generally well written and presented. This straight forward series of experiments would be a valuable resource for those considering alternatives to residual insecticide uses in such facilities.

Minor comments.

Line 44. Change “both on humans” to “both humans”

Line 49. A clear delineation is required between mosquito-borne disease and mosquito-borne pathogen. Please change “Japanese encephalitis” to “Japanese encephalitis virus”. An abbreviation can then be used, JEV, for use elsewhere – such as Line 52. There may be other instances of this throughout manuscript and so authors should check.

Line 54. In the abstract, the authors state that the use of neurotoxic insecticides are not used due to perceived health risk to animals, if this is the case, it should be stated clearly in this paragraph with supporting evidence. It is important that in papers of this kind, the "risks" associated with traditional approaches are not overstated. It would be useful to include more material here for those unaware of the mosquito control considerations in these areas.

Line 65-69. These sentences would be more suitable for inclusion in discussion rather than introduction, please delete from introduction.

Line 81-82. Could authors comment on how this application rate compares to any known registered product recommended application rate or application rates of previously published work?

Line 128. The authors should include a comment in the discussion regarding the impact of treated sheet storage in dark in laboratory compared to normal conditions in animal holding areas. Should shorter periods of residual activity be expected under “real world” conditions?

Line 142. While it would be have useful to have replicate treatment and control boxes for testing, authors should note any confounding issues that may potentially have arisen from using only a single box for the treatment or control.

Line 160. An illustration of the box is probably not required here. Inclusion of description and dimensions in the methods is sufficient.

Author Response

Response to Reviewer 1 Comments

Line 44. Change “both on humans” to “both humans”

Reply: Done

Line 49. A clear delineation is required between mosquito-borne disease and mosquito-borne pathogen. Please change “Japanese encephalitis” to “Japanese encephalitis virus”. An abbreviation can then be used, JEV, for use elsewhere – such as Line 52. There may be other instances of this throughout manuscript and so authors should check.

Reply: We changed all as suggested

Line 54. In the abstract, the authors state that the use of neurotoxic insecticides are not used due to perceived health risk to animals, if this is the case, it should be stated clearly in this paragraph with supporting evidence. It is important that in papers of this kind, the "risks" associated with traditional approaches are not overstated. It would be useful to include more material here for those unaware of the mosquito control considerations in these areas.

Reply: We added a sentence with reference “Single dermal applications of recommended doses of pyrethroids on cows resulted in the presence of residual insecticides in milk”

Line 65-69. These sentences would be more suitable for inclusion in discussion rather than introduction, please delete from introduction.

Reply: Done

Line 81-82. Could authors comment on how this application rate compares to any known registered product recommended application rate or application rates of previously published work?

Reply: We added a sentence with reference “In a previous study, a dosage of 1.3% PPF (or about 455 mg/m2) was applied for treatment of a barrier against Ae. albopictus

Line 128. The authors should include a comment in the discussion regarding the impact of treated sheet storage in dark in laboratory compared to normal conditions in animal holding areas. Should shorter periods of residual activity be expected under “real world” conditions?

Reply: We agreed and added the following sentence in discussion. “In the current study, the PPF-treated sheets were kept in a dark place during evaluation of residual effect. Hence, shorter periods of residual activity is expected if it is used under real conditions.”

Line 142. While it would be have useful to have replicate treatment and control boxes for testing, authors should note any confounding issues that may potentially have arisen from using only a single box for the treatment or control.

Reply: The objective of this experiment was not to compare the resting density of mosquitoes between the PPF-treated and control boxes, which would need more than a single box for the treatment and control. We aimed at determining the reproductivity of the blood-fed Cx. quinquefasciatus females after exposure to PPF compared with non-exposure groups. They were set on alternative nights to reduce bias.

We added ‘at the same site’ in the sentence “One control and one PPF-treated resting boxes were placed at the same site on alternative nights….”

Line 160. An illustration of the box is probably not required here. Inclusion of description and dimensions in the methods is sufficient.

Reply: We agreed to delete the illustration of the box.

Reviewer 2 Report

Title:

Consider changing to “Pyriproxyfen-Treated Polypropylene Sheets and Resting Boxes for Controlling Mosquitoes in Livestock Operations”

The authors ready too heavily upon Tables for presenting results and do not adequately describe the results in the text. It is the responsibility of the authors to interpret the results for the reader (and reviewer). The 

Minus tables, the text of the results is less than 2 pages. The authors should add more thorough description of the results using text.

Statements such as “The results of the cone bioassays using polypropylene sheets treated with PPF at concentrations of 333 and 666 mg/m2, respectively, are shown in Table 1” and “The results are given in Table 2” are not useful and should be deleted. It is much more useful and illustrative to make a declarative statement about results that references a table or figure. For example, the authors could state that “PPF concentration had a strong effect on oviposition, with 100% suppression in the 666mg/m2 group and ….333mg/m2… (Table 1)”.

The Discussion does a poor job of interpreting the results of the current study, The authors mostly rehash the results if prior work and do not critically interpret their own work. Despite seven tables of data, the results are only modestly discussed.

Abstract: Well written.

Introduction

Line 39: Replace “take up” with “acquire” 

Lines 42-43: The statement that “Zooprophylaxis … can reduce insect vector populations” is not correct. Zooprophylaxis reduces contact rates between vectors and humans by diverting bites towards non-human animals. The authors need to clarify this point.

Line 44: “animals” should be “other animals”

Line 46: “Livestock serves” should be “Livestock serve”

Lines 46-47: The statement that “Livestock serves as a plentiful source of blood, which contributes to sustaining vector populations, such as An. arabiensis and An. funestus s.l., the malaria vectors in Africa” is contrary to the authors’ previous statement about zooprophylaxis.

Line 49: Should be “Japanese encephalitis virus”

Line 53: Replace “clear” with “established”.

Line 59: The authors should avoid using terms like “normally” as in the case “PPF is normally used for controlling mosquitoes…”.  The authors can simply state that PPF is an IGR that has been used effectively to control mosquitoes in the aquatic stages” without alluding to what is normal or not normal.

Line 60: The phrase “acts to disrupt pupation and adult emergence” is not clear. PPF does not disrupt pupation. It interferes with metamorphosis.

Line 60: Delete “Moreover”

Line 66: Replace “populations” with “species”

METHODS

I am not familiar with the “cone bioassays” and am having trouble imagining the setup. The authors could include a line drawing as Figure 1 to help readers understand the method. The current figure 1 is not very useful.

Line 93: Authors should state whether or not eggs of each female were reared individually and a few details of the rearing conditions (size of pan, volume of water) since larval development is key to their study.

Line 137: Delete “feral”

Line 162: The sentence “Data were analyzed in SPSS version 22.0 (IBM Corp., Armonk, NY, USA).” should be moved to the end of the paragraph.

For their analysis I can understand the authors wishing to use standard tests such as t-test and ANOVA, however count data do not meet the assumptions of parametric statistics. I recommend that the authors begin to use Poisson regression modeling or negative binomial regression modeling for their data analysis. These analyses are a better fir for their data. Citations of a couple of very good papers are provided.

O’hara, R.B. and Kotze, D.J., 2010. Do not logtransform count data. Methods in Ecology and Evolution, 1(2), pp.118-122.

Coxe, S., West, S.G. and Aiken, L.S., 2009. The analysis of count data: A gentle introduction to Poisson regression and its alternatives. Journal of personality assessment91(2), pp.121-136.

RESULTS

See comments above.

DISCUSSION

Lines 268-280: This information is better suited to the Introduction.

Lines 282-283: Artificial resting boxes are good for sampling Anopheles but are not effective for sampling Aedes aegypti. 

TABLES

The large of tables (7) and the raw totals is very cumbersome. It is not clear why total numbers are reported. It seems as though numbers of progeny per female(mean) is more informative than the total, especially since the number of females assayed changes between initial (n=30), 3 month (n-15) and 5 month (n=15) assays.

The Table headers for tables 1-4 should be simplified, as follows, eliminating totals in favor of means. This consolidates data.

Species 

Group 

Females ovipositing (n, %) 

 Eggs / female

Larvae / female 

Pupae  / female

Adults / female 

The superscript letters meant to denote differences in treatments are very small and difficult to discern.

Figures: The authors should consider presenting some of the tabular data as figures (graphs).

Author Response

Response to Reviewer 2 Comments

Title:

Consider changing to “Pyriproxyfen-Treated Polypropylene Sheets and Resting Boxes for Controlling Mosquitoes in Livestock Operations”

 Reply: We agreed to change the title as suggested.

The authors ready too heavily upon Tables for presenting results and do not adequately describe the results in the text. It is the responsibility of the authors to interpret the results for the reader (and reviewer).

Minus tables, the text of the results is less than 2 pages. The authors should add more thorough description of the results using text.

Reply: We merged some Tables and revised data presentation in the Tables and text.

Statements such as “The results of the cone bioassays using polypropylene sheets treated with PPF at concentrations of 333 and 666 mg/m2, respectively, are shown in Table 1” and “The results are given in Table 2” are not useful and should be deleted. It is much more useful and illustrative to make a declarative statement about results that references a table or figure. For example, the authors could state that “PPF concentration had a strong effect on oviposition, with 100% suppression in the 666mg/m2 group and ….333mg/m2… (Table 1)”.

 Reply: We revised the text and tables as suggested.

The Discussion does a poor job of interpreting the results of the current study, The authors mostly rehash the results if prior work and do not critically interpret their own work. Despite seven tables of data, the results are only modestly discussed.

 Reply: We revised the discussion as suggested.

Abstract: Well written.

Reply: Thank you

Introduction

Line 39: Replace “take up” with “acquire” 

Reply: Done

Lines 42-43: The statement that “Zooprophylaxis … can reduce insect vector populations” is not correct. Zooprophylaxis reduces contact rates between vectors and humans by diverting bites towards non-human animals. The authors need to clarify this point.

Reply: We modified the sentence as below:

 Use of zooprophylaxis, which can reduces contact rates between vectors and humans by diverting bites towards non-human animals, can be integrated into vector control with chemical insecticides used in public health.

 Line 44: “animals” should be “other animals”

Reply: Done

Line 46: “Livestock serves” should be “Livestock serve”

Reply: Done

Lines 46-47: The statement that “Livestock serves as a plentiful source of blood, which contributes to sustaining vector populations, such as An. arabiensis and An. funestus s.l., the malaria vectors in Africa” is contrary to the authors’ previous statement about zooprophylaxis.

Reply: The statement about zooprophylaxis is clarified as above.

Line 49: Should be “Japanese encephalitis virus”

Reply: Done

Line 53: Replace “clear” with “established”.

Reply: Done

Line 59: The authors should avoid using terms like “normally” as in the case “PPF is normally used for controlling mosquitoes…”.  The authors can simply state that PPF is an IGR that has been used effectively to control mosquitoes in the aquatic stages” without alluding to what is normal or not normal.

Reply: We modified the sentence as below:

 Pyriproxyfen (PPF) is an insect growth regulator that has been used effectively to control mosquitoes in the aquatic stages.

Line 60: The phrase “acts to disrupt pupation and adult emergence” is not clear. PPF does not disrupt pupation. It interferes with metamorphosis.

 Reply: We deleted this and modified the sentence as above.

Line 60: Delete “Moreover”

Reply: Done

Line 66: Replace “populations” with “species”

Reply: Done

METHODS

I am not familiar with the “cone bioassays” and am having trouble imagining the setup. The authors could include a line drawing as Figure 1 to help readers understand the method. The current figure 1 is not very useful.

Reply: We added a figure for cone bioassay test.

Line 93: Authors should state whether or not eggs of each female were reared individually and a few details of the rearing conditions (size of pan, volume of water) since larval development is key to their study.

 Reply: The method of oviposition, including the size of cup and volume of water, was mentioned earlier. However, we modified the sentence to clarify.

The hatched larvae in each ovicup were reared with rabbit food (Top RabbitTM, Thailand)  until emergence of adults.

Line 137: Delete “feral”

Reply: Done

Line 162: The sentence “Data were analyzed in SPSS version 22.0 (IBM Corp., Armonk, NY, USA).” should be moved to the end of the paragraph.

Reply: Done

For their analysis I can understand the authors wishing to use standard tests such as t-test and ANOVA, however count data do not meet the assumptions of parametric statistics. I recommend that the authors begin to use Poisson regression modeling or negative binomial regression modeling for their data analysis. These analyses are a better fir for their data. Citations of a couple of very good papers are provided.

Reply: Thank you for this suggestion. We re-analyzed the data by using negative binomial regression as suggested. Most of the results are the same as our previous analysis, except some data which looks better by the new method.

O’hara, R.B. and Kotze, D.J., 2010. Do not log‐transform count data. Methods in Ecology and Evolution, 1(2), pp.118-122.

Coxe, S., West, S.G. and Aiken, L.S., 2009. The analysis of count data: A gentle introduction to Poisson regression and its alternatives. Journal of personality assessment91(2), pp.121-136.

RESULTS

See comments above.

DISCUSSION

Lines 268-280: This information is better suited to the Introduction.

Reply: We moved theses sentences to the Introduction.

Lines 282-283: Artificial resting boxes are good for sampling Anopheles but are not effective for sampling Aedes aegypti. 

Reply: Studies by Edman et al. ( J. Med. Entomol. 1998, 35, 578-583.) clearly demonstrated that artificial resting boxes are attractive to Ae. aegypti indoors and aspirating from resting boxes is an efficient and rapid method for sampling Ae. aegypti.

We have modified the sentence to clarify.

“They have proved to be useful for quickly sampling Ae. aegypti inside houses and Anopheles populations outdoors”.

TABLES

The large of tables (7) and the raw totals is very cumbersome. It is not clear why total numbers are reported. It seems as though numbers of progeny per female(mean) is more informative than the total, especially since the number of females assayed changes between initial (n=30), 3 month (n-15) and 5 month (n=15) assays.

Reply: We reduced the number of tables and modified them as suggested.

The Table headers for tables 1-4 should be simplified, as follows, eliminating totals in favor of means. This consolidates data.

Species 

Group 

Females ovipositing   (n, %) 

 Eggs / female

Larvae /   female 

Pupae  /   female

Adults /   female 

Reply: We changed all tables as suggested. 

The superscript letters meant to denote differences in treatments are very small and difficult to discern.

Reply: We increased the font of tables.

Figures: The authors should consider presenting some of the tabular data as figures (graphs).

Reply: We found difficulties to present some of data as graphs. However, after modification of Tables following your suggestions, they are informative and easy to understand.
